# Antimicrobial resistant enteric bacteria are widely distributed amongst people, animals and the environment in Tanzania

Murugan Subbiah[1,7], Mark A. Caudell[1,2,7]*, Colette Mair[3,7], Margaret A. Davis[1], Louise Matthews[3], Robert J. Quinlan[1,4], Marsha B. Quinlan [1,4], Beatus Lyimo [5], Joram Buza[5], Julius Keyyu[6] & Douglas R. Call[1,5]

Antibiotic use and bacterial transmission are responsible for the emergence, spread and persistence of antimicrobial-resistant (AR) bacteria, but their relative contribution likely differs across varying socio-economic, cultural, and ecological contexts. To better understand this interaction in a multi-cultural and resource-limited context, we examine the distribution of antimicrobial-resistant enteric bacteria from three ethnic groups in Tanzania. Household-level data ($n = 425$) was collected and bacteria isolated from people, livestock, dogs, wildlife and water sources ($n = 62,376$ isolates). The relative prevalence of different resistance phenotypes is similar across all sources. Multi-locus tandem repeat analysis ($n = 719$) and whole-genome sequencing ($n = 816$) of *Escherichia coli* demonstrate no evidence for host-population subdivision. Multivariate models show no evidence that veterinary antibiotic use increased the odds of detecting AR bacteria, whereas there is a strong association with livelihood factors related to bacterial transmission, demonstrating that to be effective, interventions need to accommodate different cultural practices and resource limitations.

[1] Paul G. Allen School for Global Animal Health, Washington State University, Pullman, WA, USA. [2] Food and Agriculture Organization of the United Nations, Nairobi, Kenya. [3] Boyd Orr Centre for Population and Ecosystem Health, Institute of Biodiversity, Animal Health and Comparative Medicine, University of Glasgow, Glasgow, UK. [4] Department of Anthropology, Washington State University, Pullman, WA, USA. [5] Nelson Mandela African Institution of Science and Technology, Arusha, Tanzania. [6] Tanzania Wildlife Research Institute, Arusha, Tanzania. [7] These authors contributed equally: Murugan Subbiah, Mark A. Caudell, Colette Mair. *email: mcaudell@wsu.edu

By 2030 the rising demand for livestock products in low- and middle-income countries (LMICs) will increase global antimicrobial consumption in the agriculture sector by almost 70%[1]. This prospect has motivated calls for increased antimicrobial stewardship in agriculture, including the reduction and elimination of antimicrobials as growth promoters and unnecessary prophylaxis. In theory, these changes should reduce the magnitude of antibiotic use, leading to a reduction in the abundance of antimicrobial-resistant bacteria and related transfer of resistance traits to zoonotic pathogens, and a reduction in transmission of antimicrobial-resistant bacteria through the food chain and environment[2–9].

Besides the use of antimicrobials, transmission of genetic traits and of resistant bacteria is a key component of the antimicrobial-resistance challenge. Because tracking specific transmission events is rarely practical, transmission is often evaluated indirectly by characterizing the distribution of different bacterial phenotypes and genotypes within and between different host organisms. This comparative data can then be evaluated in the context of antimicrobial use, sociocultural and economic variables to identify correlations and potential risk factors for bacterial transmission. Importantly, risk factors can vary considerably depending on the sociocultural and economic context. For example, Mathers et al.[10,11] documented a limited degree of phenotypic and genotypic similarity between multidrug-resistant *Salmonella* collected from people and livestock in Scotland, and similar patterns were found for ESBL/AmpC *Escherichia coli* (*E. coli*) isolates in the Netherlands[12], and for resistant *Salmonella* isolates in the United States[13]. The lack of overlap is consistent with limited direct contact between food-animal reservoirs and the population in general, and probably limited zoonotic transmission via food and water. In contrast, similar studies conducted in LMICs document overlap between people and their animals. In Uganda, where there are extensive interactions between people and livestock, there is also considerable genotypic similarity between resistant *Salmonella* isolates from people and their animals[14]. Similar

overlap has been reported for ESBL-producing *E. coli* in Tanzania[15]. While a study conducted within the Netherlands found overlap between livestock (pigs) and people, the overlap was dependent on intensity of contact and was more pronounced in farming communities[16].

In these cases, transmission likely arises through food- and water-borne transmission, as well as transmission through environmental contact. Finding overlap between bacteria from people and animals is important, but to develop effective interventions we must determine what risk factors are most correlated with potential transmission opportunities. Evidence of overlap also stresses that successful efforts to reduce the burden of antimicrobial-resistant bacteria in lower-income settings must simultaneously addresses risk factors for both people and livestock.

For the current study, we use antimicrobial-resistant Gram-negative, lactose-fermenting enteric bacteria (*Enterobacteriaceae*) as a model for how zoonotic bacteria might spread to different hosts. We identify risk factors for carriage of antimicrobial-resistant bacteria under the assumption that colonization with resistant bacteria is associated with previous microbial transmission, particularly in cases where antibiotic exposure is unlikely (e.g., wildlife). To assess how different risk factors might contribute to the prevalence of antimicrobial-resistant bacteria in different contexts, our study includes three ethnic groups; Maasai pastoralists, Arusha agropastoralists, and Chagga highland farmers (see Fig. 1). While inhabiting similar areas in northern Tanzania, the three groups vary considerably across sociocultural and economic dimensions that have been proposed as risk factors for antimicrobial resistance, including patterns of antimicrobial use, animal husbandry practices, proximity to urban environments, and hygiene and sanitation practices[17]. Our goals include documenting the magnitude of phenotypic (*Enterobacteriaceae*) and genetic (*E. coli*) overlap between isolates collected from people and animals (domestic and wildlife), and identifying the household-level practices (e.g., antibiotic use, animal husbandry) that may contribute to a higher prevalence of antibiotic-resistant

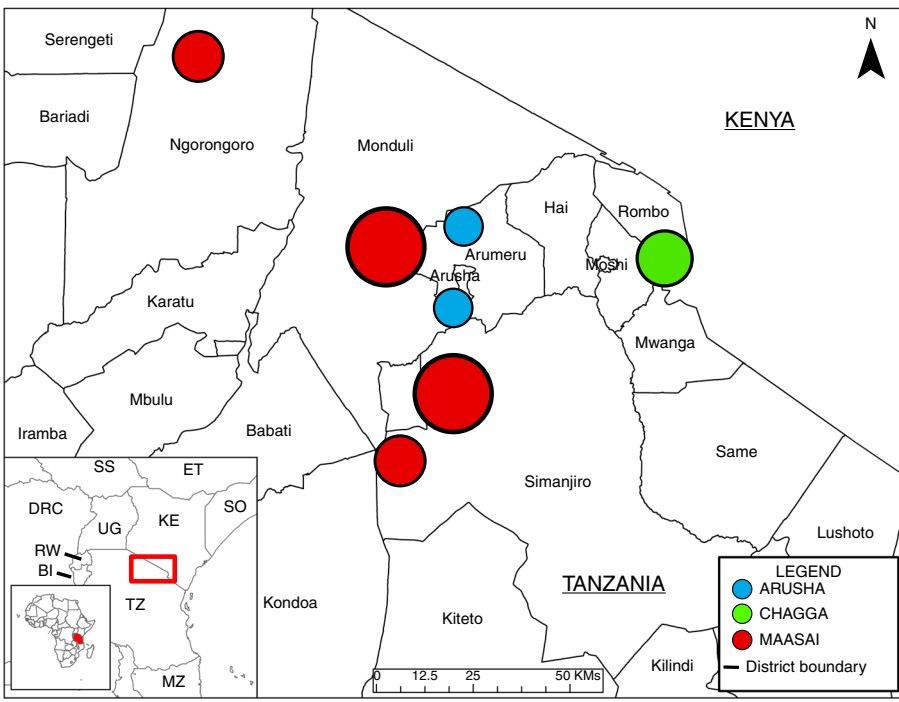

**Fig. 1 Map of study area and areas surveyed.** Arusha = 103 households, Chagga = 101 households, and Maasai = 201 households. Maps were created using ArcGIS software by Esri. The base map is sourced from Esri and modified in ArGIS Pro. "Light Gray Canvas" [basemap] https://www.arcgis.com/home/item.html?id=ee8678f599f64ec0a8ffbfd5c429c896. 30 October 2019.

**Table 1 Presence and absence data.**

| Antibiotic | G+/P+ | G−/P+ | G+/P− | G−/P− | Sensitivity[a] | Specificity[b] |
|---|---|---|---|---|---|---|
| Ampicillin | 164 | 8 | 27 | 539 | 0.86 | 0.99 |
| Ceftazidime | 7 | 1 | 10 | 724 | 0.41 | >0.99 |
| Chloramphenicol | 12 | 3 | 46 | 677 | 0.21 | >0.99 |
| Ciprofloxacin | 5 | 11 | 44 | 678 | 0.10[c] | 0.98 |
| Kanamycin | 0 | 4 | 0 | 734 | n.a.[d] | >0.99 |
| Streptomycin | 165 | 23 | 55 | 489 | 0.75 | 0.96 |
| Sulfamethoxazole | 186 | 34 | 23 | 495 | 0.89 | 0.94 |
| Tetracycline | 237 | 42 | 19 | 440 | 0.93 | 0.91 |
| Trimethoprim | 153 | 35 | 11 | 539 | 0.93 | 0.94 |
| Amp + Str + Sul + Tet + Tri[e] | 905 | 142 | 135 | 2502 | 0.87 | 0.95 |
| Amp + Sul + Tet + Tri[f] | 740 | 119 | 80 | 2013 | 0.90 | 0.94 |

Comparison of the presence (+) or absence (−) of antimicrobial-resistance genotypes (G) and phenotypes (P), and associated estimates of diagnostic sensitivity and specificity of the phenotype results relative to the presence of a corresponding antimicrobial-resistance gene
[a]Diagnostic sensitivity is the proportion of isolates that were antibiotic resistant based on a breakpoint assay and that had a corresponding antimicrobial-resistance gene based on whole-genome sequencing (i.e., correctly identifies a true positive)
[b]Diagnostic specificity is the proportion of isolates that were antimicrobial-susceptible based on a breakpoint assay and that had no corresponding antimicrobial-resistance gene based on whole-genome sequencing (i.e., correctly identifies a true negative)
[c]Diagnostic sensitivity for ciprofloxacin resistance is very low even when considering the limited sample size. This is likely due to resistance being conveyed by chromosomal mutations in contrast to the presence of specific resistance genes that would normally be identified using ResFinder[49] software
[d]Not applicable due to zero value n the G+/P+cell
[e]Pooled analysis for ampicillin, streptomycin, sulfamethoxazole, tetracycline and trimethoprim tests
[f]Pooled analysis for ampicillin, sulfamethoxazole, tetracycline, and trimethoprim tests

*E. coli* at the household level. Our results show widespread overlap of resistance across species and multivariate models provide no evidence that veterinary antibiotic use increased the odds of detecting AR bacteria whereas there is a strong association with livelihood factors related to bacterial transmission. These results highlight that AMR intervention philosophies must adapt to situations where increased risk of transmission following selection from antimicrobial use is overwhelmed by the general transmission of bacteria across hosts and the environment. To identify the practices promoting this transmission across cultures will require input from interdisciplinary teams from the natural and social sciences.

## Results

**Methods assessment.** For this study, we employed methodologies for selecting and characterizing bacteria from fecal samples that were conducive to moderately high throughput assays. We isolated Gram-negative, lactose-fermenting bacteria by testing samples with MacConkey agar plates. Our goal was to select up to 48 isolates per sample, where samples consisted of pooled feces from up to three individuals of each host at each household. As reported earlier[18], this sample size of 48 isolates provided >50% probability of detecting at least one isolate from a stool sample that was resistant to a given antibiotic when the true prevalence of that resistance phenotype was only 2%.

While our goal was to focus on *Escherichia coli*, simple selection from MacConkey agar lacks specificity. We conducted whole-genome sequencing for a subset of 1317 isolates and found that for people, 90.7% of the isolates were *E. coli*[18], but the diversity of recovered isolates varied for cattle (71.1% *E. coli*), chickens (92.5%), dogs (70.3%), and sheep and goats (91.4%). Consequently, the antimicrobial resistance data presented below includes mixed species that we refer to as *Enterobacteriaceae*, but the genetic analysis was restricted to *E. coli* isolates that were confirmed by either sequencing or by PCR[19].

We assessed antimicrobial resistance using a breakpoint assay. For this assay, single isolates of bacteria were transferred to MacConkey agar plates containing fixed concentrations of individual antibiotics. Isolates that formed colonies were considered resistant, while isolates that failed to grow were considered susceptible. To assess the reliability of this technique we compared genome sequences from a subset of 732–742 *E. coli* isolates from

this study with their expected antimicrobial resistance phenotypes (Table 1). Restricting this analysis to the five most prevalent resistance phenotypes (the other resistance phenotypes were represented by only 4 to 16 isolates), the percentage of isolates having identifiable antibiotic resistance genes and a corresponding resistance phenotype was 87% (diagnostic sensitivity). The percentage of isolates having no identifiable antimicrobial-resistance genes and that were typed as susceptible to the five antibiotics was 95% (diagnostics specificity). Streptomycin resistance, for which there is no Clinical and Laboratory Standard Institute recommended breakpoint[20], had the greatest error with 75% sensitivity and 95% specificity. This reduced sensitivity likely reflects our use of a higher-than-necessary concentration of streptomycin in our agar plates, but this error should be distributed randomly across all samples. When streptomycin data were removed, diagnostic sensitivity and specificity increased to 90% and 94%, respectively. We attribute the 6–10% discrepancy for diagnostic sensitivity and specificity to (1) true error, (2) isolates having intermediate resistance, which could be interpreted as resistant or susceptible using agar plates with fixed concentrations of antibiotics, (3) isolates having characteristics that confer intrinsic resistance, and (4) isolates having nonfunctional antimicrobial-resistance genes[21]. For the analysis presented below, we have assumed that these errors are randomly distributed across sources.

**Similar distribution of resistant bacteria across groups.** Our first finding was that the relative distribution of antimicrobial-resistant bacteria is similar across groups, but there were differences in the prevalence between groups. From 315 households, a total of 43,691 Gram-negative, lactose-fermenting bacteria (Chagga = 11,755; Maasai = 24,635; Arusha = 7301) were isolated from cattle, small stock (goats and sheep), dogs and chickens (Table 2). An additional 11,470 isolates were isolated from human stool samples and were reported elsewhere[18], but these findings were used here as a point of reference. Generally, the mean prevalence of antimicrobial-resistant bacteria across livestock types was higher for Maasai and Arusha compared to Chagga households (Fig. 2). In addition, resistance to ampicillin, streptomycin, sulfamethoxazole, trimethoprim, and tetracycline across livestock types was higher (>35% in Arusha, >30% in Maasai, and >10% in Chagga) compared to ceftazidime, chloramphenicol, ciprofloxacin, and kanamycin. An exception to this

**Table 2 Number of households and isolates from animals, water, and wildlife.**

|  | Chagga | | Maasai | | Arusha | |
|---|---|---|---|---|---|---|
|  | # of isolates | # of HHs | # of isolates | # of HHs | # of isolates | # of HHs |
| Cattle | 3696 | 63 | 6371 | 118 | 1892 | 43 |
| Chickens | 3117 | 62 | 4872 | 96 | 2234 | 56 |
| Dogs | 624 | 13 | 5910 | 110 | 1435 | 34 |
| Sheep/goat | 4318 | 70 | 7482 | 140 | 1740 | 38 |
| People[a] | 4608 | 85 | 4274 | 79 | 2405 | 58 |
| Water[b] | 110 | n.a. | 427 | n.a. | 1397 | n.a. |
| Wildlife[c] | 5464 | | | | | |

HHs households
[a]Human isolate data published in ref. [18]
[b]Water was collected from sources within or near Chagga, Maasai and Arusha communities[23]
[d]Wildlife isolates were collected opportunistically and without reference to specific communities

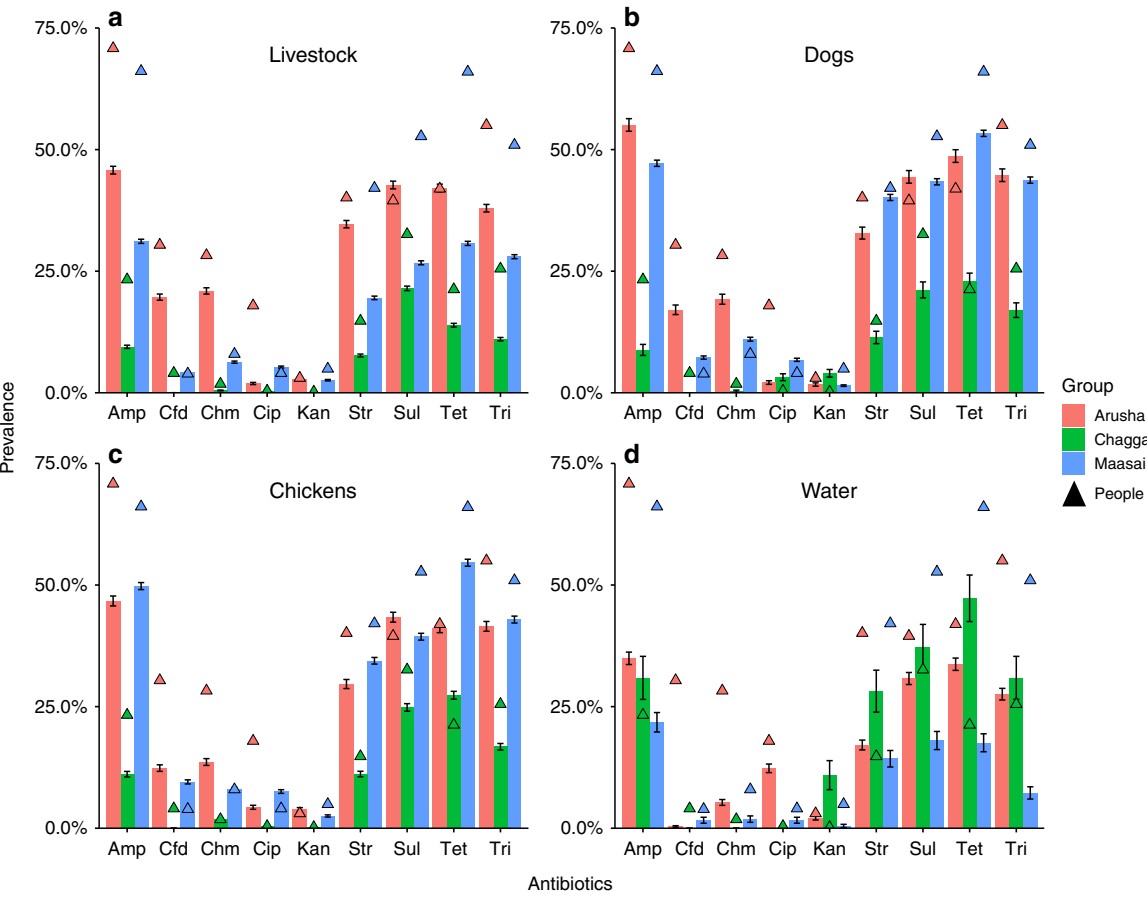

**Fig. 2 Prevalence of antimicrobial resistant bacteria in people, animals, and water.** Bacteria isolated from fecal samples collected from Maasai, Arusha, and Chagga people ($n = 11,287$ isolates) and animals ($n = 43,691$ isolates) and water samples ($n = 1934$ isolates). Antibiotics included amp (a mpicillin), cfd (ceftazidime), chm (chloramphenicol), cip (ciprofloxacin), and kan (kanamycin), str (streptomycin), sul (sulfamethoxazole), tet (tetracycline), tri (trimethoprim). Error bars are 95% standard errors.

pattern was resistance to ceftazidime and chloramphenicol in Arusha where ~20% of isolates were resistant. A comparable distribution and prevalence of antimicrobial-resistant bacteria was found for opportunistically collected wildlife fecal samples (Fig. 3). In some cases, the prevalence of resistant bacteria was higher in waters from Chagga and Arusha communities compared to Maasai. The prevalence of the nine antimicrobial-resistance phenotypes was highly correlated between bacterial isolates from households and bacteria from wildlife ($r^2 = 0.91$; multivariate analyses of variance (MANOVA), $P < 0.001$).

**E. coli genotypes randomly distributed across hosts**. Our second finding was that *E. coli* genotypes are distributed randomly across communities and host sources. Multilocus variable-number of tandem repeats (MLVA) are highly mutable genetic markers that are useful for distinguishing between closely related strains of bacteria[22]. For this MLVA comparison, a haplotype was defined as the combination of different alleles detected for five different tandem-repeat loci, and for the current analysis there were 388 unique haplotypes (combinations of MLVA alleles). MLVA analysis for 324 *E. coli* isolated from cattle, small stock,

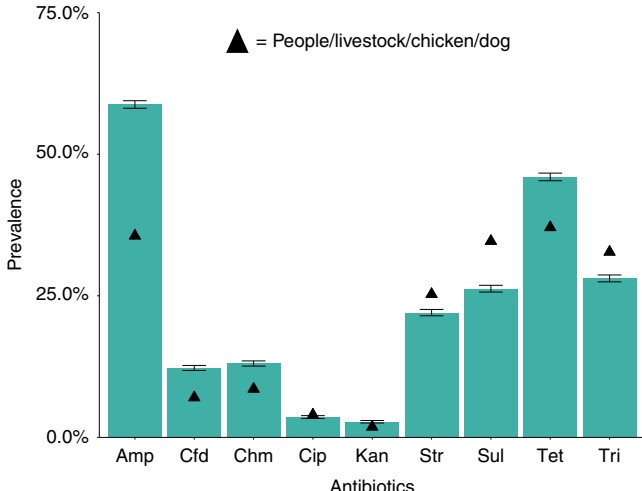

**Fig. 3 Prevalence of antimicrobial-resistant bacteria in wildlife.** Bacteria isolated from fecal samples collected from wildlife ($n = 5464$ isolates) compared to mean prevalence of resistance from people/livestock/chicken/dog combined (triangles; $n = 54,978$ isolates). Wildlife fecal samples were opportunistically collected from wildebeest (Connochaetes taurinus), zebra (Equus quagga), impala (Aepyceros melampus), giraffe (Giraffa camelopardalis), elephant (Loxodonta africana), gazelle (Eudorcas thomsonii), dik-dik (Madokua kirkii), and buffalo (Syncerus caffer). Antibiotics included amp (ampicillin), cfd (ceftazidime), chm (chloramphenicol), cip (ciprofloxacin), and kan (kanamycin), str (streptomycin), sul (sulfamethoxazole), tet (tetracycline), tri (trimethoprim). Error bars are 95% standard errors.

chickens, dogs, people, wildlife and water showed that ≥90% of haplotypes were shared between these hosts. In addition, over 98% of *E. coli* haplotypes from livestock were shared between *E. coli* isolated from water and wildlife (Table 3). These findings indicate that there was little distinctiveness for the haplotypes of *E. coli* that were collected from different sources when compared to the diversity of haplotypes found within sources. This was also consistent with minimum spanning tree from MLVA data (Fig. 4 & Supplementary Fig. 1 for comparisons between hosts), and with MLST-based (Fig. 5) and core-genome-based phylogenetic trees (Supplementary Fig. 2) that were constructed from *E. coli* sequence data ($n = 84$ isolates) that originated from eight Massai households (see methods). We identified a number of older-antimicrobial resistance genes from the subset of sequenced *E. coli* (Supplementary Data 1)[23–25] Sixteen isolates (2%) harbored $bla_{CTX-M-15}$ and one isolate harbored $bla_{CMY-42}$.

**Prevalence associated with general transmission of bacteria.** Our third set of findings derived from multilevel logistic models, clustered at the household level, that were used to assess the livelihood factors associated with detection of antimicrobial resistant bacteria (see Identification of risk factors in Methods for variable descriptions). Our primary outcome was whether an isolate was resistant to an antibiotic(s) (1) or was susceptible (0). Models were specified for the five most prevalent resistance phenotypes (ampicillin, streptomycin, sulfamethoxazole, tetracycline and trimethoprim) and for multi-drug resistant phenotypes (MDR), which we define as resistance to three or more of these five antibiotics. We restricted our results to those variables that were statistically significant for at least three phenotypes, and results are presented as odds ratios where odds above 1 indicate increased odds of resistance and <1 decreased odds.

*Livestock, pooled analysis*: as households increased the number of people with whom they exchanged livestock there was an increasing probability of detecting antimicrobial-resistant bacteria in livestock across all ethnic groups, with large increases in odds for ampicillin (odds ratio (OR): 3.5, 95% confidence interval (CI): 2.05–5.95), streptomycin (OR: 2.19, CI: 1.52–3.15), sulfamethoxazole (OR: 2.08, CI: 1.41–3.05), tetracycline (OR: 1.91, CI: 1.35–2.71), trimethoprim (OR: 2.7, CI: 1.68–4.36), and MDR (OR: 2.93, CI: 1.92–4.47) (see Supplementary Data 2 for full model results and Identification of risk factors in Methods for variable descriptions). Increasing use of livestock markets was also associated with large increases in odds of detecting bacteria resistant to ampicillin (OR: 2.83, CI: 1.79–4.47), streptomycin (OR: 2.58, CI: 1.69–3.94), sulfamethoxazole (OR: 1.95, CI: 1.30–2.93), tetracycline (OR: 1.95, CI: 1.30–2.91), trimethoprim (OR: 1.87, CI: 1.22–2.86), and MDR (OR: 2.98, CI: 1.81–4.88). Increasing distance between households and major urban centers (i.e., towns of Arusha and Moshi) decreased the odds of detecting bacteria resistant to ampicillin (OR: 0.4, CI: 0.29–0.57), streptomycin (OR: 0.66, CI: 0.49–0.89), tetracycline (OR: 0.59, CI: 0.44–0.78), trimethoprim (OR: 0.48, CI: 0.35–0.65), and MDR (OR: 0.44, CI: 0.29–0.66).

*Arusha livestock only*: the only livelihood factor significantly related to antimicrobial resistance in Arusha livestock ($n = 58$ households) was the number of livestock exchange partners with which a household interacted (see Supplementary Data 3 for full model results). As the numbers of these partners increased, the odds of detecting resistant bacteria increased for ampicillin (OR: 2.88, CI: 1.31–6.30), streptomycin (OR: 3.05, CI: 1.26–7.37), sulfamethoxazole (OR: 2.29, CI: 1.01–5.23), trimethoprim (OR: 2.37, CI: 1.05–5.33), and MDR (OR: 3.21, CI: 1.29–8.03).

*Chagga livestock only*: for the Chagga ($n = 72$), households that purchased more livestock in the last year had livestock with reduced odds of having *E. coli* resistant to streptomycin (OR: 0.39, CI: 0.21–0.70), tetracycline (OR: 0.63, CI: 0.41–0.97) and trimethoprim (OR: 0.43, CI: 0.26–0.72) (see Supplementary Data 4 for full model results). Households further away from the city of Moshi exhibited lower odds of resistance to ampicillin (OR: 0.43, CI: 0.23–0.82), trimethoprim (OR: 0.38, CI: 0.21–0.70), and MDR (OR: 0.29, CI: 0.12–0.68).

*Maasai livestock only*: for Maasai livestock ($n = 100$), the odds of harboring antimicrobial-resistant *E. coli* were significantly lower for households with higher rates of veterinary antibiotic use (see Supplementary Data 5 for full model results). Households keeping more antibiotics/syringes at home and reporting higher use rates had lower odds of resistance to ampicillin (OR: 0.26, CI: 0.10–0.64), streptomycin (OR: 0.39, CI: 0.21–0.74), sulfamethoxazole (OR: 0.37, CI: 0.19–0.73), trimethoprim (OR: 0.42, CI: 0.20–0.87) and MDR (OR: 0.23, CI: 0.10–0.53). Having a larger number of livestock exchange partners increased the odds of resistance to all antibiotics, including ampicillin (OR: 2.95, CI: 1.73–5.03), streptomycin (OR: 2.02, CI: 1.50–2.73), sulfamethoxazole (OR: 2.05, CI: 1.44–2.91), tetracycline (OR: 1.83, CI: 1.31–2.56), trimethoprim (OR: 2.39, CI: 1.59–3.60), and MDR (OR: 2.66, CI: 1.85–3.83). Likewise, as the number of livestock a person managed for somebody outside the household increased so did the odds of resistance to streptomycin (OR:1.42, CI: 1.05–1.93), sulfamethoxazole (OR: 1.47, CI: 1.08–2.00), trimethoprim (OR: 1.73, CI: 1.21–2.49), and MDR (OR: 1.52, CI: 1.02–2.28). Households that utilized more livestock markets exhibited higher odds of resistance to ampicillin (OR: 2.38, CI: 1.04–5.45), streptomycin (OR: 2.09, CI: 1.17–3.74), and MDR (OR: 2.3, CI: 1.11–4.77).

*Chickens and Dogs Pooled:* no livelihood factors were associated with the odds of detecting antimicrobial-resistant bacteria from chickens across a majority of antibiotics (see Supplementary Data 6 for full model). The only factor related to a majority of resistance phenotypes was ethnic affiliation, with

**Table 3 The distribution of *E. coli* haplotypes within and between different sources.**

|  | Within sources % variation | Between sources % variation | Sample size *n* vs *n* |
|---|---|---|---|
| People vs. water | 91.05 | 8.95 | 99, 136 |
| People vs. wildlife | 90.96 | 9.04 | 99, 90 |
| People vs. chickens | 93.7 | 6.4 | 99, 63 |
| People vs. dogs | 91.5 | 8.5 | 99, 94 |
| People vs. (chickens, dogs) | 92.8 | 7.2 | 99, 157 |
| People vs. livestock | 93.5 | 6.5 | 99, 68 |
| Livestock vs. (chickens, dogs) | 92.6 | 7.4 | 68, 157 |
| Livestock vs. water | 98.3 | 1.7 | 68, 136 |
| Livestock vs. wildlife | 98.2 | 1.8 | 68, 90 |
| Livestock vs. (people, dogs, chickens) | 94.3 | 5.7 | 68, 256 |
| Water vs. wildlife | 98.7 | 1.3 | 136, 90 |
| Water vs. (people, dogs, chickens, livestock, wildlife) | 92.8 | 7.2 | 136, 414 |
| Wildlife vs. (people, dogs, chickens, livestock) | 96.1 | 3.8 | 90, 324 |

Results are from the analysis of MLVA data using AMOVA. Sample sizes are people (*N* = 99), livestock (*n* = 68), dog (*n* = 94), chicken (*n* = 63), wildlife (*n* = 90), water (*n* = 136)

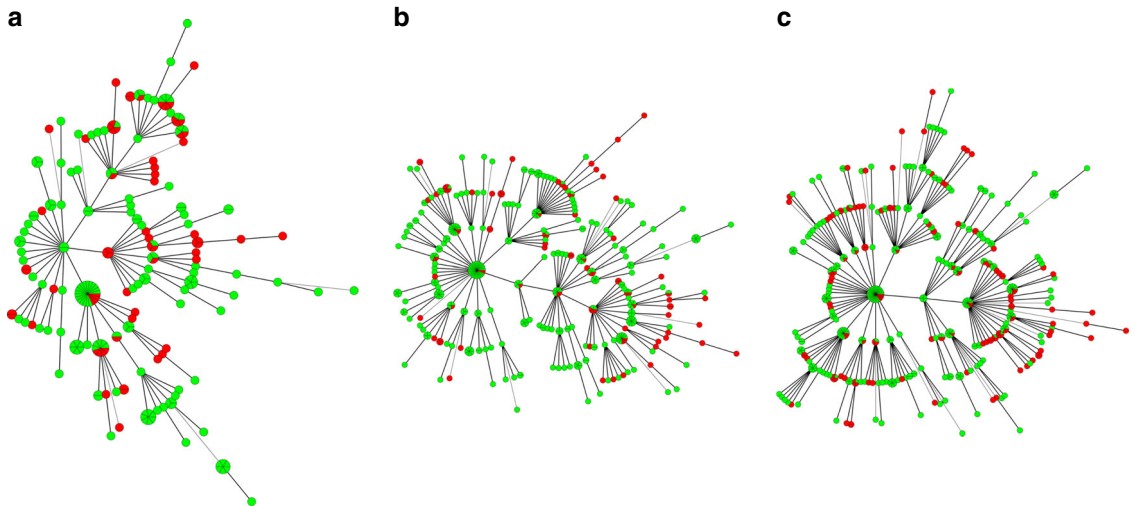

**Fig. 4 Minimum spanning tree for E. coli MLVA haplotypes.** Each circle or pie-slice represents a single E. coli isolate from people or animal. Most of the isolates differed by a single locus (solid lines) and no host-specific clustering was apparent. Sample sizes are people (*n* = 99), livestock (*n* = 68), dog (*n* = 94), chicken (*n* = 63), wildlife (*n* = 90), water (*n* = 136).

Chagga chickens exhibiting much lower odds of resistance to ampicillin (OR: 0.02, CI: 0.20–0.71), streptomycin (OR: 0.13, CI: 0.03–0.62), trimethoprim (OR: 0.18, CI: 0.04–0.78), and MDR (OR: 0.03, CI: 0.01–0.21) after controlling for livelihood variables related to selection and transmission (consistent with Supplementary Fig. 2C). For dogs, the odds of harboring resistant *E. coli* were significantly lower for households that reported boiling milk, including for ampicillin (OR: 0.3, CI: 0.13–0.72), streptomycin (OR: 0.43, CI: 0.19–0.97) and tetracycline (OR: 0.4, CI: 0.17–0.93) (see Supplementary Data 7 for full model). In households that took more steps to avoid disease in their animals (e.g., graze sick animals separately, spray/dip livestock), dogs exhibited decreased odds to ampicillin (OR: 0.62, CI: 0.4–0.96), sulfamethoxazole (OR: 0.53, CI: 0.32–0.88), and trimethoprim (OR: 0.58, CI: 0.38–0.88). As households purchased more livestock, odds of resistant bacteria from dogs decreased for streptomycin (OR: 0.59, CI: 0.35–0.99), sulfamethoxazole (OR: 0.46, CI: 0.25–0.84) and trimethoprim (OR: 0.49, CI: 0.29–0.82). As distance from urban centers increased there was a large *increase* in the odds of detecting resistant bacteria from dogs to all antibiotics, including ampicillin (OR: 2.06, CI: 1.14–3.72), streptomycin (OR: 3.2, CI: 1.82–5.62), sulfamethoxazole (OR: 3.46, CI: 1.77–6.77), tetracycline (OR: 2.47, CI: 1.39–4.38), and trimethoprim (OR: 3.11, CI:

1.76–5.49). After controlling for the proposed risk factors for antimicrobial resistance, dogs owned by Chagga households had much lower odds of resistance to ampicillin (OR: 0.01, CI: 0.0–0.13), streptomycin (OR: 0.09, CI: 0.01–0.71), sulfamethoxazole (OR: 0.06, CI: 0.00–0.67), trimethoprim (OR: 0.06, CI: 0.01–0.53), and MDR (OR: 0.01, CI: 0.0–0.11).

**Discussion**

We examined the distribution of antimicrobial-resistant enteric bacteria from animals, people and water in northern Tanzania, and we examined the probability of detecting these bacteria relative to livelihood factors from three culturally diverse ethnic groups. Our findings highlight four main points. First, enteric bacteria in northern Tanzania exhibit a high prevalence of antimicrobial resistance with over 50% of the ≈50,000 isolates from domestic animals, wildlife, and water sources displaying resistance to at least one antibiotic. In addition, resistance to five specific antibiotics (ampicillin, streptomycin, sulfamethoxazole, tetracycline, and trimethoprim) was consistently higher than resistance to other antimicrobials tested across animals and environmental samples, a result that we also documented for people in northern Tanzania[18]. Second, comparison of MLVA

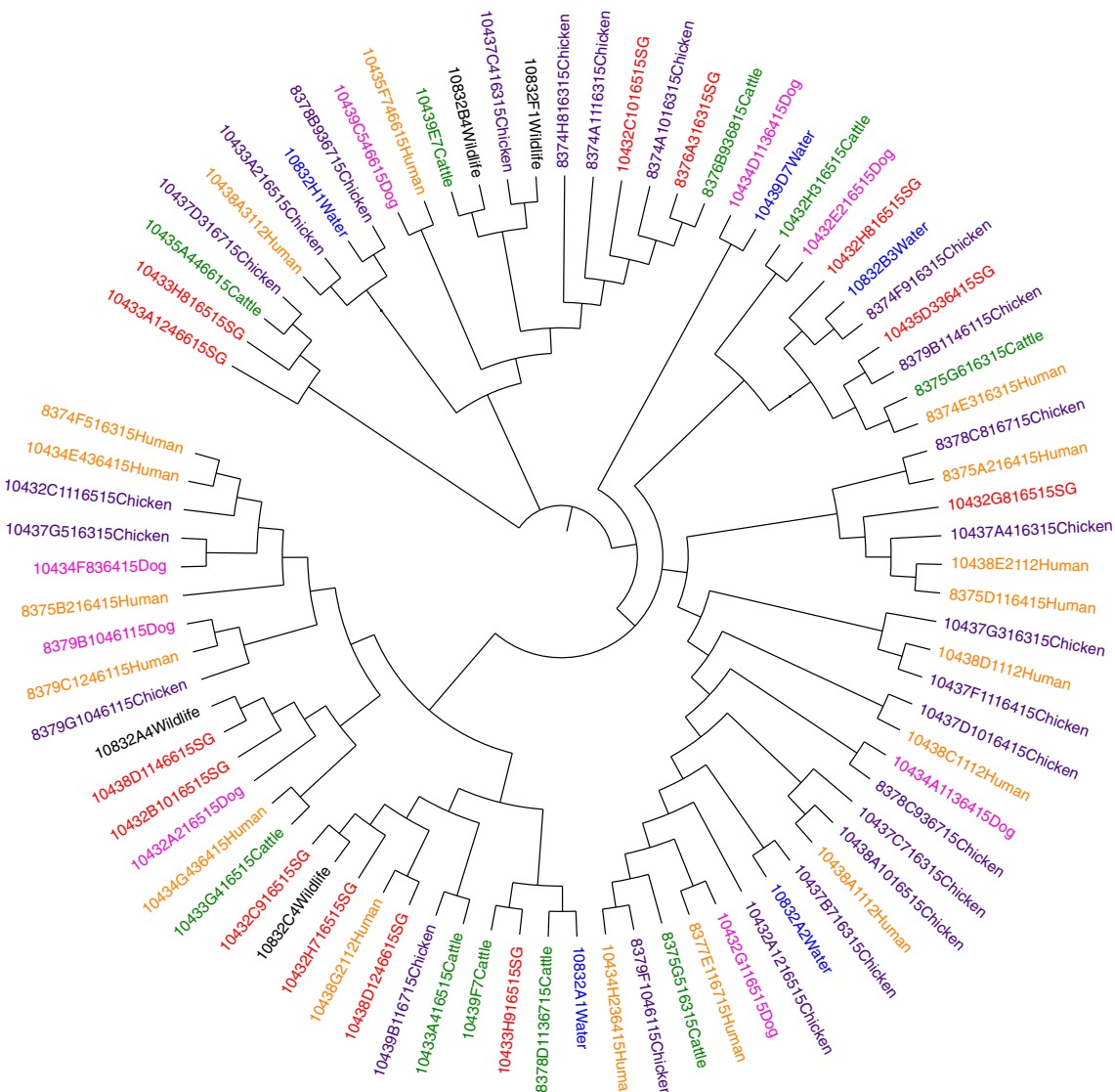

**Fig. 5 Phylogenetic tree derived from 81 E. coli isolates.** Isolates were collected from eight selected Maasai households and also include wildlife and waters isolates. No clustering of isolates was apparent based on host species or households. Labels show barcode id, house id, year of collection, and host name (e.g. 10435D5466Human; 10435D5—barcode id, 466—household id, and Human—host species).

data demonstrated that *E. coli* isolates from our sampled populations are genetically diverse and are widely distributed across all sources. This lack of host-association was validated for the subset of *E. coli* that were characterized genetically (MLST typing and whole-genome comparisons). Over 80% of isolates exhibited MLVA haplotypes that were also present in people, livestock, wildlife, and water sources. These results are consistent with studies documenting high degrees of overlap in low-income countries, including Uganda[14] and Tanzania[15], but are in contrast to most studies conducted in high-income countries[10–13]. Consistent with the phylogenic analysis, our third finding was that livelihood factors that could promote the general transmission of bacteria, such as visiting markets, sharing water sources, and distance to major urban centers, were more strongly associated with prevalence of antimicrobial-resistant bacteria relative to factors related to antibiotic use. That is, within these three communities, any specific effect of antibiotic use on the transmission of resistant bacteria appears to be overshadowed by livelihood factors that promote the transmission of bacteria generally. The importance of these livelihood factors is also consistent with our fourth finding that the prevalence of

antimicrobial-resistant bacteria was not uniformly distributed across ethnic groups. Indeed, samples from Maasai pastoralist- and Arusha agropastoralist-households exhibited prevalence levels ~two-fold greater than Chagga highland farmers. These cross-cultural differences are consistent with Maasai and Arusha subsistence practices that include keeping larger herds and drawing from a wider range of sources (i.e., markets, livestock exchange partners) compared to the Chagga[17].

Cross-cultural differences in livelihood factors related to antimicrobial resistance highlight the necessity of incorporating interdisciplinary social science perspectives to understand the complexity of behaviors that may be driving the emergence, spread, and persistence of these bacteria. Developing this understanding will be particularly essential for designing interventions within LMICs given the diversity of livelihoods that continue to be pursued (pastoralism, agro-pastoralism, farming, paid labor). In addition, limited input from the professional health sector within many LMICs ensures that animal treatment practices are informed by traditional ethnomedical belief systems. These systems impact risk communication strategies, which must consider, for example, how cultural understandings of contagion

impact comprehension of antimicrobial resistance[26]. Understanding of ethnomedical belief systems can also give rise to novel intervention approaches. Our intervention among Maasai, for example, has shown how promoting milk pasteurization, over the common method of boiling, can increase the frequency of milk heat-treatment, thereby reducing a major risk of bacterial transmission to people[27,28].

A more nuanced understanding of antimicrobial resistance as a sociocultural phenomenon is needed given, as our results suggest, seemingly related livelihood factors between different communities could be inversely correlated with the likelihood of detecting antimicrobial resistance. For example, practices that promote animal movement in Maasai and Arusha communities (i.e., livestock exchange relationships and the number of markets visited) were associated with large increases (≈200–350%) in the odds of detecting bacteria from livestock that were resistant to one of five antibiotics and a multidrug-resistance phenotype. These results are consistent with studies finding that the introduction of new animals to herds increases the likelihood of introducing novel strains of bacteria[29]. In contrast, livestock movement in Chagga highland farmers, as defined by the number of animals purchased in the last year, was associated with reduced odds (≈40–60%) of detecting antimicrobial-resistant bacteria from livestock. This finding may reflect how livestock are integrated into Chagga households relative to Maasai and Arusha households. Most Chagga households have very few cattle (mean ≈ 1 animal) and they keep cattle primarily for production of dairy products. Consequently, when Chagga households report having purchased cows in the last year (as asked through our questionnaire) they are more likely to purchase older animals that can start producing milk sooner. This is important because young cattle normally harbor a much higher prevalence of antimicrobial-resistant enteric bacteria compared with older animals[30,31]. If this pattern holds in Tanzania, purchasing presumptively older animals would lead to a reduced overall prevalence of antimicrobial-resistant bacteria simply as a function of animal age. In contrast, Maasai and Arusha households are more likely to purchase younger cattle as an investment for later return after a few years of grazing the animal. As a result, livestock purchasing patterns in the Maasai and Arusha may ensure that younger animals are introduced to the herd thereby increasing overall prevalence of antimicrobial-resistant bacteria independent of other livelihood factors.

Cross-cultural differences may also explain different patterns of antimicrobial resistance for bacteria collected from dogs. Dogs from households that boiled milk had lower odds (≈60–70%) of having enteric bacteria that were resistant to ampicillin, streptomycin, and tetracycline. Earlier results from this project showed that consuming raw milk is a major risk factor for detecting antimicrobial-resistant bacteria in people[18]. Maasai households, given limited use of toilets [≈20% of households[17]], are likely to leave human fecal waste unprotected and dogs may be scavenging this waste and consequently acquiring antimicrobial-resistant bacteria from people. Many households also feed milk directly to dogs as a porridge mix with crushed corn (once per day, biased to the rainy season), which could lead to direct transmission from milk as we surmise occurs for people[18]. Another result linked to cultural differences may be the unexpected relationship between increasing distance to urban centers and the odds of dogs harboring more antimicrobial-resistance bacteria. This association may reflect a correlation between larger herds that would not be otherwise sustainable closer to urban areas, and this in turn might may lead to dogs receiving more milk. It is also possible that there are more waste scavenging opportunities at outlying households given that the likelihood of improved waste management systems (e.g., toilets, sewers, etc.) decreases with distance to urban centers.

Based on our analysis, there was no evidence that veterinary antibiotic use increased the odds of detecting antimicrobial-resistant bacteria. At one level, this is not totally unexpected. The primary veterinary antibiotic used in these communities is an injectable tetracycline [oxytetracycline[17]. When injected, most of this drug is eventually excreted in urine rather than in feces and consequently, there probably isn't much direct selection pressure in vivo from this application[32]. Some excreted antibiotics can enrich populations of resistant bacteria in exposed soils [e.g., florfenicol[32], but if there is a clay component to the soil it is likely that oxytetracycline will adsorb to the soil and will be mostly unavailable[33]. It is also possible that our methods for collecting antibiotic use data (i.e., household inventories and self-report) did not accurately reflect long-term antibiotic use patterns. Indeed, the Chagga had much lower levels of resistance overall and this may reflect a cultural history of less antibiotic use due to smaller herd sizes and better herd health, and greater reliance on professional veterinary services. The possible influence of these historical patterns highlights the importance of applying a mixed-methods approach to understand distributions of antimicrobial-resistant bacteria in the present.

One remarkable consistency from this study was that regardless of host, environmental source or cultural group, there was a relatively high prevalence of bacteria that were resistant to ampicillin, streptomycin, sulfamethoxazole, tetracycline, and trimethoprim. This was true for the current study, an earlier report focused on people[18], and other studies from food animals from the Arusha peri-urban area[34–36]. All of these studies characterized antimicrobial resistance using the same methodology whereby bacteria were grown on agar plates with a fixed concentration of antibiotics. This is a low-cost means of assessing antibiotic resistance, but it provides no estimate for the variance in the magnitude of resistance for individual isolates as would be possible with disc-diffusion or microdilution assays. This method is also subject to interpretation issues if the population of bacteria being tested are not representative of the strains that are used to develop international standards for antibiotic breakpoint concentrations (although this does not preclude relative comparisons within studies). Our analysis of whole-genome sequences suggests that our methods provide a high degree of diagnostic sensitivity and specificity for detection of antimicrobial resistance. Nevertheless, it is possible that our breakpoint methodology produces systematic artefacts. For example, if a breakpoint concentration is too high (as suggested here for streptomycin, Table 1), the prevalence of resistant organisms will appear low. Such artefacts could produce a similar distribution of resistant organisms across different hosts (e.g., high prevalence of resistance to Amp, Str, Sul, Tet and Tri). To examine this possibility, we isolated and characterized *E. coli* from a very different source (municipal wastewater in Pullman, WA) using the same methods described for the current study. The pattern of resistance ($n = 381$ isolates) was quite different from our findings in Tanzania, including considerably lower prevalence overall for four out of the five most common resistance phenotypes (see Supplementary Fig. 3). Consequently, it is unlikely that our reported findings for the prevalence of antimicrobial-resistant bacteria are due to methodological artifacts.

Documentation of the widespread similarity of antimicrobial-resistant bacteria between animals, people, and the environment represents an important step forward in the promotion of evidence-based approaches to address antimicrobial-resistance on a global scale. On a regional level, stark differences in the prevalence of antimicrobial-resistant bacteria documented in this study (i.e., higher prevalence) compared to studies from high-income countries (i.e., lower prevalence) suggests that broad intervention philosophies must adapt to situations where

**Table 4 Livelihood dimension differences between the Maasai, Arusha, and Chagga.**

| Characteristics | Maasai | | | Arusha | | | Chagga | | |
|---|---|---|---|---|---|---|---|---|---|
| | Mean | Median | IQR | Mean | Median | IQR | Mean | Median | IQR |
| Mean number of livestock (cattle and shoats) | 291 | 147 | 251 | 15 | 3.5 | 15 | 5 | 5 | 6 |
| Some formal education | 30% | 0 | 1 | 67% | 1 | 1 | 92% | 1 | 0 |
| Self-administer veterinary antibiotics | 98% | 1 | 0 | 42% | 0 | 1 | 2% | 0 | 0 |
| Seek professional veterinary services | 24% | 0 | 0 | 30% | 0 | 1 | 92% | 1 | 0 |

Numbers are reported as averages and are rounded. IQR is the interquartile range

increased risk of transmission following selection from anti-microbial use is overwhelmed by the general transmission of bacteria across hosts and the environment. While prudent-use approaches are often the centerpiece of intervention efforts[37–39], as well as the motivation behind efforts to understand anti-microbial resistance, our study suggests that higher priority should be given to improving sanitation, hygiene, and healthcare infrastructures than to prudent-use strategies, especially in resource limited settings[40]. This assignment of intervention priorities is especially critical given the limited availability of public health resources and competing demands at both inter-national and national levels. As this study highlights, assigning priorities and subsequent development of targeted strategies will require cross-cultural investigations developed and implemented by interdisciplinary teams from the natural and social sciences.

## Methods

**Study design**. A team of veterinarians, livestock extension officers, microbiologists, ecologists, epidemiologists, social scientists, and local community members nego-tiated and coordinated the research design, planning, logistics, project imple-mentation, data management, and analysis. We used a mixed-methods strategy that combined qualitative and quantitative data collection beginning with 20 for-mal, qualitative, key informant and focus group interviews among Maasai and Chagga livestock owners across a range of communities in 2012. We interviewed livestock extension officers and veterinarians in different communities to deter-mine common practices in different areas and the course of professional veterinary training in Tanzania. We used data from initial interviews to refine our survey instruments. We observed livestock management and veterinary care in multiple Maasai and Chagga households in 2012 including use of veterinary antimicrobials, chemical dips, and traditional treatments. We observed necropsy of recently deceased animals, slaughter and butchering, milking and milk handling, breeding, birthing, branding, grazing and fodder provision, and castration. Informal inter-views were conducted during the course of direct observation, including con-versations that recounted people's health experiences, those of their livestock, and additional detail on the circumstances surrounding illness events. Iterative quali-tative interviewing helped to add or modify existing survey instruments as different ethnic groups were studied and as lab results of fecal and milk samples suggested new items for inclusion. We visited multiple animal drug shops in Monduli, Simanjiro, Arusha City, and Moshi districts, and interviewed attendants about veterinary antibiotic sales and recommended usage. We held three community meetings after quantitative data collection was complete in those communities. Meetings served as focus groups and opportunities to report our preliminary results. These meetings also allowed us to discuss public health solutions relevant to the study communities. Unless otherwise indicated, our ethnographic description of livestock management and veterinary practices draws on these materials.

A 200-item questionnaire was administered across 13 villages between March 2012 and July 2015, with village selection based upon consultation with local research assistants and officials and in consideration of other research projects in the area[17]. Across villages, 425 households were randomly selected using census lists provided by village- or ward-executive offices. Focal villages with sampling season and year are provided in Supplementary Table 1. Research assistants fluent in English, Swahili and Maa or Chagga were trained and employed (Maasai assistants in Maasai and Arusha villages and Chagga assistants in Chagga villages) for data collection and to facilitate participation. Surveys were conducted in *Maa* among Maasai and in *Kiswahili* among Chagga and Arusha. Participant consent was verbally attained given high rates of illiteracy within the study populations. Verbal consent was documented in a separate data file indicating informant ID numbers, agreement to participate, and payment/receipt of informant fees. The study was reviewed and approved by the Washington State University (IRB #12355) and Tanzania National Institute for Medical Research institutional review boards, and a research permit was issued by the Tanzania Commission for Science and Technology (permit 2012-151). A research permit for collecting voided wildlife

fecal samples were issued by the Tanzania Wildlife Research Institute (No. 2015–127-ER-2012-51). Other ethical review for animal use was not required because fecal samples were collected from voided materials.

**Study groups Arusha Chagga Maasai**. Study groups were selected as they varied across a spectrum of subsistence systems, animal healthcare practices, and devel-opment indicators (e.g., education, hygiene and sanitation) proposed as drivers of AMR. Many Maasai people in this region continue to inhabit remote geographic areas where they combine maize and bean farming with the tending of large, free-ranging herds of livestock that congregate at communal waterholes that are used by people, livestock, and wildlife[41]. Access to human and animal healthcare remains limited and many Maasai use antibiotics without seeing a healthcare professional. Most Maasai households administer antibiotics to their animals without oversight from veterinary services[17]. Arusha people inhabit peri-urban areas around the city of Arusha, a major urban area (population ~ 400,000), combining cultivation of maize and banana with the tending of smaller herds that may access water at neighborhood standpipes[17,42]. Close proximity to the city allows greater access to modern education and healthcare systems, including access to professional veter-inary services[17]. This proximity also means Arusha face many of the health issues typical of rapidly urbanizing populations, including those associated with high population densities and poor sanitation services[43]. Chagga peoples live near Moshi town, another major urban center (population ~ 200,000) where they practice banana cultivation, keep home gardens and tend small herds that are confined/tethered and brought fodder and water[44]. Compared to the other groups, the Chagga rely the most on professional healthcare services and almost always seek livestock health professionals to diagnose and treat their animals[17] (Table 4).

**Fecal sampling and laboratory methods**. Fresh voided fecal samples from people, livestock, chickens, and dogs were collected at the households and placed into sterile plastic bags. Fresh voided wildlife fecal samples were collected opportunis-tically from surrounding areas. Bacterial isolates from water sources were obtained from a previous study[23]. Fecal samples were kept in a portable refrigerator and were transported to a laboratory at Nelson-Mandela African Institution of Science and Technology (NM-AIST) in Arusha, Tanzania. Fecal samples were serially diluted and plated onto MacConkey agar and were then incubated overnight at 37 °C. Up to 48 Gram-negative, lactose-fermenting colonies (per sample) were picked and inoculated individually into wells (containing 150 µl LB broth) of 96-well assay plates. After inoculation, the plates were incubated overnight at 37 °C and glycerol (20% final concentration) was added before storing at −80 °C. For transportation, archived plates were used to prepare 40 µl of fresh overnight cul-tures in 96-well plates where it was allowed to air-dry before sealing and shipping to Washington State University under ambient conditions. Once received, cultures were revived with the addition of 150 µL of LB broth into each well and incubated overnight at 37 °C.

For the MLVA assay, the identity of presumptive *E. coli* isolates was confirmed by PCR testing for the presence of *uidA* following the methods of Bej et al., 1991[19] with modification. Briefly, overnight cultures of each isolate (1-mL LB broth) were centrifuged, pelleted and then suspended in 1-mL sterile water and boiled in a water bath for 15 min. The boiled suspensions were briefly centrifuged at 500 rpm for 2 min and supernatants were used as template for PCR. Template (1 µl) was added to a reaction mixture with primers *uidA*-specific primers (F: 5′ TGGTAATTACCGACGAAAACGGC 3′; R: 5′ ACGCGTGGTTACAGTCTTGCG 3′). Cycles included denaturation at 95 °C (3 min) followed by 35 cycles of 95 °C (30 sec), 58 °C for 30 s and 72 °C for 1 min followed by final extension at 72 °C for 10 min.

A previously described breakpoint assay[45] was used to test for susceptibility to nine antibiotics. Briefly, isolates were allowed to thaw and were transferred using a 96-pin replicator into a new 96-well plate with fresh media for overnight incubation at 37 °C. Isolates were then manually transferred using the pin-replicator onto MacConkey agar plates (150 mm) made with a fixed concentration of a single antibiotic (ampicillin, amp, 32 µg/ml; ceftazidime, cef, 8 µg/ml; chloramphenicol, chm, 32 µg/ml; ciprofloxacin, cip, 4 µg/ml; kanamycin, kan, 32 µg/ml; streptomycin, str, 120 µg/ml; sulfamethoxazole, sul, 512 µg/ml; tetracycline, tet, 16 µg/ml; trimethoprim, tri, 8 µg/ml) and incubated overnight at 37 °C. Isolates were considered resistant if well-formed colonies were visible, but

susceptible if poorly grown colonies (presumptive intermediate) or no growth were observed. After incubation, the numbers of resistant and susceptible bacteria were enumerated and the average prevalence of resistant bacteria per antimicrobial per sample per household was estimated for summary figures (e.g., 24 out of 48 isolates were resistant to ampicillin, average ampicillin resistance prevalence is 24/48 = 50%).

**Genotyping of *E. coli*.** To characterize genetic similarities between *E. coli* strains from different sources, 719 confirmed *E. coli* isolates were selected and genotyped using a MLVA assay[22,46]. At the household level, only one randomly selected isolate per unique combination of antimicrobial resistance per fecal sample was included. Overall, these isolates represented all available hosts (animals and people) from each household from 11 villages (3 Arusha, 4 Chagga, and 4 Maasai) and 44 households (4 households per village that had the maximum number of hosts represented). In addition, isolates from wildlife feces (mixed species), and waters[23] were also included. Overnight cultures (1-mL LB broth) were centrifuged, pelleted and then suspended in 1-mL sterile water and boiled in a water bath for 15 min. The boiled suspensions were briefly centrifuged at 500 rpm for 2 min and supernatants were used as template for PCR.

Five primer sets were used to amplify five MLVA loci. Forward primers from each primer set were tagged with fluorescent reporters (Supplementary Table 2) and involved two independent multiplex PCR reactions (mix 1 or 2). PCR reactions were assembled with commercial reagents (Qiagen Multiplex PCR-kit, Hilden, Germany) as detailed by the manufacturer. DNA template was prepared as described for *uidA* PCR. Template (1 μL) was added to the PCR reaction for a total volume of 25 μL. Both multiplex PCR reactions were processed by using a thermocycler (Bio-Rad, Hercules, CA, USA) with the following conditions: 95 °C for 15 min, then 30 cycles of 94 °C for 30 s, 60 °C for 90 s and 72 °C for 90 s, followed by a hold at 72 °C for 10 min. PCR amplicons were then diluted 1:100 in water, and 5 μL of the diluted PCR-amplicon was added to 14.7 μL of formamide (Ambion Inc., Austin, TX, USA) and 0.3 μL of Genscan 600 LIZ internal size standard (Applied Biosystems, Foster City, CA, USA). This mix was then subjected to capillary electrophoresis using an ABI 3730 DNA Analyzer (Thermo Fisher, Waltham, MA USA) at the Molecular Biology and Genomics Core, Washington State University, Pullman, WA. Capillary electrophoresis was run at 66 °C with POP7 polymer (Thermo Fisher) for 20 min at 15 kV. MLVA data were imported to Bionumerics (version 6.6) program as character data and MST clusters were identified[22]. Variation in the distribution of MLVA haplotypes within and between host species was evaluated by using analysis of molecular variances (AMOVA, R-version 3.5.1; ade4; popr.amova).

**Multi-locus sequence typing of *E. coli*.** DNA extractions from 854 *E. coli* isolates were sent to a service center (MicrobesNG, University of Birmingham, UK; https://microbesng.uk/) for whole-genome sequencing using an Illumina platform. Isolates

were originally selected from ten Maasai households, water and wildlife samples to address a separate question about the degree of genetic similarity relative to household social and physiographic connectivity, and thus this sample was not a random sample from the study population. Nevertheless, the sequence data provided an opportunity to validate MLVA findings with respect to the distribution of different *E. coli* strains amongst hosts. The sequencing service provided assembled contigs from each genome using an in-house bioinformatics pipeline. This included trimming with Trimmomatics and quality control evaluation using in-house scripts combined with Samtools, BedTools and bwa-mem (MicrobesNG, University of Birmingham, UK; https://microbesng.uk/).

Bioinformatic utilities from the Center for Genomic Epidemiology (CGE) were used to identify the sequence types (STs, MLST 1.8 – MultiLocus Sequence Typing[47]), to detect plasmid sequences (PlasmidFinder 1.3[48]), and to detect antimicrobial-resistance genes (ResFinder 3.0[49]). The batch upload mode was used to upload multiple sequences at the same time (a total of 816 *E. coli* whole-genome sequences), and default settings were used (e.g., a 90% sequence similarity threshold was used to identify antimicrobial-resistance genes).

For construction of a core-genome phylogenetic model, sequences from seven housekeeping genes (*adk*, *fumc*, *gybr*, *icd*, *mdh*, *pura*, and *recA*) for each *E. coli* strain (*n* = 816) and *E. coli* K12 (substrain MG1655; NCBI accession number: NC_CP032667) genome were extracted, concatenated and aligned as a single FASTA file and uploaded to CGE (CSI Phylogeny 1.4) to identify sequence variants (single-nucleotide polymorphisms, SNPs). Out of 816 *E. coli* sequences, one sequence per host per household from each clade was randomly picked (*n* = 81 isolates) and uploaded to MEGA6 software[50] to create a maximum-likelihood phylogenetic tree.

To assess the resolution of the ST-based phylogenetic tree, a phylogenetic tree was also constructed based on whole-genome alignment of these 81 sequences. Command-line-based *parsnp* and *gingr* programs from a core-genome analysis suite Harvest were used to construct the tree[51]. Output from *gingr* was extracted as a *Newick* file and was uploaded to iTOL (Interactive Tree of Life, an online based tool)[51,52] to display and manage the phylogenetic tree. The clades of ST- and whole-genome-based phylogenetic trees were compared visually.

**Identification of risk factors.** From our household surveys, we compiled sets of variables that have the potential to be associated with transmission and/or selection for antimicrobial-resistant bacteria in livestock, chicken, and dogs (Table 5 and Supplementary Data 8 for statistics by ethnic group). Variables were selected based upon results from reviews of antimicrobial resistance in low- and middle-income countries (e.g., the WHO Global Action Plan[2]) and from our ethnographic work (i.e., recurrent practices we observed that likely impacted transmission/selection)[17].

For the livestock models, we first pooled ethnic groups together and then presented group-specific models. For dog and chicken models, we only used a pooled analysis due to sample size restrictions (10–20 households for one or more ethnic groups). For these latter models we included ethnic affiliation as a control

---

### Table 5 Description of variables entered into multivariate models.

| Variable | Description |
|---|---|
| Boil Milk (1 = Yes 0 = No) | Whether a household normally boiled their drinking milk before consumption |
| Distance from HH to nearest urban center | Geodesic distance in kilometers between household and the nearest urban center. Nearest urban center was Arusha or Moshi |
| HH health care visits | The number of times any member of the household went to visit a clinic in the last six months |
| HH antibiotic use | A scale of antibiotic use potential that include the number of antibiotics, syringes, and recalled use of antibiotics in last month |
| HH vaccination use | The number of diseases all livestock had been vaccinated against. Importantly, this did not indicate whether an entire herd had been vaccinated for a particular disease |
| Vet services used | The number of veterinary services used including government veterinarians, private veterinarians, community animal health workers, agrovets shopkeepers, and animal health laboratory workers |
| Livestock exchange partners | The number of unique individuals a household had exchanged cattle with in the last year |
| Livestock in and out of home | The number of livestock (cattle, sheep, goats) that moved in and out of the household on a daily basis. The herd would leave to graze/water in the morning and return near sunset |
| Livestock purchased | The number of livestock (cattle, sheep, goats) purchased in the last year |
| Markets used | The number of markets a household used to buy and sell livestock (cattle, sheep, goats) |
| Outside livestock managed | The number of livestock (cattle, sheep, goats) that a person managed for someone outside the household. |
| Scale of urbanity | A scale of urbanity including whether the household had any form of electricity, radio, tv, refrigerator, motorcycle, vehicle, and number of cellphones |
| Steps taken to avoid disease | The number of steps taken by households to avoid diseases in their herds including, keeping calves separate, making an isolation shed, grazing sick cattle separately, supplementing feed, vaccinating, and spraying |
| Toilet (1 = Yes 0 = No) | Whether the household used a flush/pit toilet |
| Total animals at home | The total number of animals kept at the household including cattle, sheep, goats, donkeys, chickens, pigs, ducks |
| Waterholes used | The number of waterholes NORMALLY used by a household throughout the year |
| Water source shared with animals | A variable indicating whether livestock, wildlife, and people shared the same water source |

variable and tested for interaction effects between significant risk factors and ethnic affiliation. Results were summarized as odds ratios (OR), with odds ratios >1.0 indicating a higher odd of detecting antimicrobial-resistant bacteria and <1.0 one indicating a lower odd of exhibiting resistance. Variables were reported as Z-scores so that changes in odds ratios should be interpreted as changes relative to changes in standard deviation. To minimize attention to potentially spurious correlations, we restricted our inferences to variables that were significantly associated with resistance to three or more of the six possible antimicrobial-resistance phenotypes. Model fit was assess using McKelvey and Zavoina Pseudo $R^2$, a goodness of fit measure that is based on variance decomposition of the estimated logits and has been recommended as measure for logistic multilevel models[53–55]. Most values of McKelvey and Zavoina Pseudo $R^2$ were 0.3 or above, indicating good model fit. See Supplementary Data 9 for McKelvey and Zavoina Pseudo $r^2$ values for estimations using both the fixed and random effects, the fixed-effects only, and the intraclass correlations.

**Reporting summary**. Further information on research design is available in the Nature Research Reporting Summary linked to this article.

## Data availability

Sequence data that support the findings of this study have been deposited in GenBank at National Center for Biotechnology Information [repository name "BioProject ID PRJNA578301; Genome submission SUB6444306"] with the accession codes SAMN13068707 through SAMN1068790. The socioeconomic data that support findings of this survey are available at figshare with identifier doi: 10.6084/m9.figshare.10185077.

## Code availability

No custom code was used in the analyses.

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

## Acknowledgements

This study was funded in part by the US National Science Foundation (DEB 1216040), the UK Biotechnology and Biological Sciences Research Council (grant numbers BB/K01126X/1 and BB/L018926/1), the UK Medical Research Council (grant number MC_UU_12014/9). We are grateful for the invaluable assistance of many people, including Chagga, Arusha, and Maasai study participants, the village chairmen of focal villages (Isaya Rumas, Godfrey Naisikye, Lemuta Naisikye, Imma Laiser, Willium Kanunga, Imani Baraka, Paul Sangre, Joseph Tarimo, Joseph Temo, Rigobert Tarimo, and Deogratius Mshanga), and the Nelson Mandela African Institution of Science and Technology, which provided invaluable logistical support.

## Author contributions

L.M., R.J.Q., M.B.Q. and D.R.C. designed the study. M.A.C., M.S., R.J.Q., M.B.Q., D.R.C. and B.L. planned and completed data collection. M.S., M.A.D. and B.L. did the laboratory analysis and M.A.C. did the statistical analyses. M.A.C., C.M., M.S., L.M. and D.R.C. wrote the first draft of the article and the paper was revised and critically reviewed by B.L., J.B., J.K., M.A.D., M.B.Q. and R.J.Q.

## Competing interests

The authors declare no competing interests.
