## [Peer Review File · Nature Communications]

Reviewers' Comments:

Reviewer #1:

Remarks to the Author:

General

The paper does not do what the authors claim has to be done, i.e. study transmission. As I agree what has to be done I am disappointed. I would suggest rewording the goals (major revision) or the journal to reject the paper.

Antibiotic use and transmission of bacteria are not two factors that influence the problems with AMR at the same level. Whereas use of antibiotics is necessary for antibiotic resistance to emerge, all other occurrences of colonization with AMR bacteria are associated with the transmission of bacteria with AMR. Antibiotic use comes in the picture again as it could promote transmission of bacteria with AMR and/or reduce the transmission of bacteria without AMR. In practice under conditions that antibiotics are not used bacteria with AMR seem to transmit equally well than the same bacterium without the AMR (Broens et al., 2012a; Broens et al., 2012b; Huijbers et al., 2016). This is probably the case because AMR seems to have low costs and even these low costs are compensated by compensatory mutations.

L55 Transmission determines strategy. Agree but in this paper transmission is not studied.

L70 Agree but in this paper transmission is not studied.

L78 Why assume that transmission is measured by colonisation and not give arguments whether that is indeed so. In that way you can assume anything and thus draw wrong conclusions. To make myself very clear, according to me: transmission is not measured by colonisation! Transmission is the transfer of (micro-)organism from one host (the infectious one) to the other host (the recipient one). Only by taking into account exposure one can estimate transmission. This can be done using for example by using #cases/#susceptibles as dependent variable and putting #infectious/#total number in the offset (Biemans, de Jong, and Bijma, 2017; Velthuis et al., 2002). Here you look only at recipient which can have very different exposure and even for wildlife exposure could also come from infectious individual that are treated with antibiotics.

L84 In this paper they look only at overlap and not at transmission.

L371 No mention anymore of transmission ?

L389 As difference in exposure is not taken into account this estimates and the conclusion maybe very wrong. To see this be reminded that as this analysis is about prevalence (dependent variable), transmission and use of antibiotics (explanatory variables) and all these will be clustered in space and time and thus spurious results may quickly follow. Really there are four transmission rate parameters when taking into account antibiotic use: S (recipient) and I (seeder) both not treated, S treated I not, S not I treated, both S and I treated. Thus effect of antibiotic use is part of the transmission parameter estimate not a separate effect.

Given the remarks above it will not be surprising that I do agree to base advices on this study. The study is interesting as it finds results about the distribution of AMR bacteria that is different from what is found in developing countries.

Biemans, F., de Jong, M.C.M. and Bijma, P., 2017. A model to estimate effects of SNPs on host susceptibility and infectivity for an endemic infectious disease. *Genetics Selection Evolution* 49.
Broens, E.M., Espinosa-Gongora, C., Graat, E.A.M., Vendrig, N., Van der Wolf, P.J., Guardabassi, L., Butaye, P., Nielsen, J.P., de Jong, M.C.M. and Van de Giessen, A.W., 2012a. Longitudinal study on transmission of MRSA CC398 within pig herds. *Bmc Veterinary Research* 8.
Broens, E.M., Graat, E.A.M., van de Giessen, A.W., Broekhuizen-Stins, M.J. and de Jong, M.C.M., 2012b. Quantification of transmission of livestock-associated methicillin resistant *Staphylococcus*

aureus in pigs. *Veterinary Microbiology* 155, 381-388.

Huijbers, P.M.C., Graat, E.A.M., van Hoek, A., Veenman, C., de Jong, M.C.M. and van Duijkeren, E., 2016. Transmission dynamics of extended-spectrum beta-lactamase and AmpC beta-lactamase-producing *Escherichia coli* in a broiler flock without antibiotic use. *Preventive Veterinary Medicine* 131, 12-19.

Velthuis, A.G.J., de Jong, M.C.M., Stockhofe, N., Vermeulen, T.M.M. and Kamp, E.M., 2002. Transmission of *Actinobacillus pleuropneumoniae* in pigs is characterized by variation in infectivity. *Epidemiology and Infection* 129, 203-214.

Mart C.M. de Jong

Reviewer #2:

Remarks to the Author:

General Remarks. On initial reading this looked like an interesting article attempting to examine "one health AMR" in Tanzania - a LMIC region. The sample size is impressive and at first glance so is some of the analysis. However, the paper suffers from many serious deficiencies. Firstly, the sampling criteria whilst is balanced lacks rationale, for example, how were the subset of *E. coli* strain chosen for WGS? Secondly, Assessing crude resistance is something that should be avoided as it does not differentiate intrinsic resistance from acquired and many of the bacteria on the screening plate we no doubt environmental bacteria or commensals. Therefore, it is very hard to ascribe any importance to the crude resistance data. Thirdly, we are told that a subset of *E. coli* were subject to WGS yet the analysis we have is very superficial - what were the ARGs and did you look for virulence factors? Fourthly, the mathematical modelling data is interesting but these to be found in the supplementary data only - in fact the choice of figures and tables for the main paper needs to be reassessed. Fifth, much of the analysis and narrative shows a degree of naivety with incorrect statements or conclusions that do align with the data - for example, the choice of antibiotics and not examining ESBLs is very strange. Also, the authors were surprised that samples showed resistance to ampicillin, streptomycin, sulfamethoxazole, tetracycline and trimethoprim - yet resistance mediated by all of these genes can be house in class 1 integrons which are very common.

In addition to the above, the use of English is careless with many incorrect sentences and typos throughout.

To conclude, the sample size is impressive but the analysis is poor and the study looks as though it was rushed.

Reviewer #3:

Remarks to the Author:

This is an interesting paper describing detailed analysis of enormous amount of information, collected in very interesting setting.

I was not convinced with one of the major messages of the study: Antibiotic use does not contribute to prevalence of resistance, it is mostly livelihood factors (in the setting studied). While it is very convincing the livelihood factors are important, there is very little supporting evidence in the study that antibiotic use is not important as a facilitator of transmission in a high transmission setting. I suggest to tone down the statement regarding antibiotic effect, and just state that under the

condition studied - no effect was found. You may want to discuss that this lack of effect may be due to close linearity with livelihood conditions, or because in high transmission setting this effect is difficult to ascertain, or because resistance to "old" antibiotics were studied, where the importance of antibiotic pressure has already reached its peak.

Line 48, up to 48 isolates were picked from..., and latter average resistance per sample was calculated. - this results in an unconventional measure of resistance which is the product of how many samples had a resistant bacteria (among those picked randomly), and what was the abundance of resistant organisms in the sample. This measure and the interpretation needs to be discussed.

Analysis - of the similarity of mechanisms of resistance in isolates that were sequenced, will add an important dimension to the paper.

Specific comments:

line 51 repeats the same message as the sentence before. please revise.

Table 2: please add the number of samples.

Reviewers' comments:

Reviewer #1 (Remarks to the Author):

General

The paper does not do what the authors claim has to be done, i.e. study transmission. As I agree what has to be done I am disappointed. I would suggest rewording the goals (major revision) or the journal to reject the paper. Antibiotic use and transmission of bacteria are not two factors that influence the problems with AMR at the same level. Whereas use of antibiotics is necessary for antibiotic resistance to emerge, all other occurrences of colonization with AMR bacteria are associated with the transmission of bacteria with AMR. Antibiotic use comes in the picture again as it could promote transmission of bacteria with AMR and/or reduce the transmission of bacteria without AMR. In practice under conditions that antibiotics are not used bacteria with AMR seem to transmit equally well than the same bacterium without the AMR (Broens et al., 2012a; Broens et al., 2012b; Huijbers et al., 2016). This is probably the case because AMR seems to have low costs and even these low costs are compensated by compensatory mutations.

>>>Our manuscript title probably set inappropriate expectations and Reviewer 1 was particularly emphatic about this point with respect to interpreting the narrative. Our goal was not to study transmission *per se*, but to identify relationships between livelihood factors and the distribution of antimicrobial-resistant enteric bacteria. The factors that we identified are consistent with transmission being a significant underlying mechanism that is responsible for the distribution of antimicrobial-resistant bacteria in the study area. Our inference about transmission is based on multiple consistent outcomes, which is a valid epidemiological methodology. Indeed, measuring actual transmission events for non-disease-causing bacterium would be extraordinarily difficult. We have modified the title and narrative to better convey the fact that our conclusions concerning the importance of transmission are inference-based rather than literal estimates of exposure and transmission. We have also strengthened our narrative about the importance of antibiotic use in the broader sense (and we agree with the general biological points made by the reviewer)<<<

L55 Transmission determines strategy. Agree but in this paper transmission is not studied. <<<please see response to first comment>>>

L70 Agree but in this paper transmission is not studied. <<<please see response to first comment>>>

L78 Why assume that transmission is measured by colonisation and not give arguments whether that is indeed so. In that way you can assume anything and thus draw wrong conclusions. To make myself very clear, according to me: transmission is not measured by colonisation! Transmission is the transfer of (micro-)organism from one host (the infectious one) to the other host (the recipient one). Only by taking into account exposure one can estimate transmission. This can be done using for example by using #cases/#susceptibles as dependent variable and putting #infectious/#total number in the offset (Biemans, de Jong, and Bijma, 2017; Velthuis et al., 2002). Here you look only at recipient which can have very different exposure and even for wildlife exposure could also come from infectious individual that are treated with antibiotics.

>>>please see response to first comment; we repeatedly identified livelihood factors associated with a higher probability of harboring antimicrobial-resistant bacteria and these factors are consistent with increased risk of bacterial transmission (e.g., using livestock markets and sharing watering holes). Population genetic analysis was also consistent with what would be expected if transmission is common – that is, little to no population subdivision<<<

L84 In this paper they look only at overlap and not at transmission. <<<please see response to first comment>>>

L371 No mention anymore of transmission? L389 As difference in exposure is not taken in to account this estimates and the conclusion maybe very wrong. To see this be reminded that as this analysis is about prevalence (dependent variable), transmission and use of antibiotics (explanatory variables) and all these

will be clustered in space and time and thus spurious results may quickly follow. Really there are four transmission rate parameters when taking into account antibiotic use: S (recipient) and I (seeder) both not treated, S treated I not, S not I treated, both S and I treated. Thus effect of antibiotic use is part of the transmission parameter estimate not a separate effect. Given the remarks above it will not be surprising that I do agree to base advices on this study. The study is interesting as it finds results about the distribution of AMR bacteria that is different from what is found in developing countries.

>>>We do not discount the importance of antibiotic use (see first paragraph of introduction), but use per se is not predictive for the system we studied. Antibiotic use is the ultimate selective force that results in emergence and proliferation of resistant strains. It is unclear, however, the degree to which spatial and temporal discontinuities separate these selective “events” from where antimicrobial-resistant bacteria are found. Our analysis failed to identify an antibiotic-use signature while finding multiple variables associated with transmission risk (this was true for a separate publication that focused on human isolates only, Lancet Planetary Health 2018). The reviewer correctly notes that statistical associations of this nature can be spurious. Our manuscript makes this point (last paragraph of the methods) with the following sentence, “To minimize attention to potentially spurious correlations, we restricted our inferences to variables that were significantly associated with resistance to three or more of the six possible antimicrobial-resistance phenotypes.” We also highlighted this point in the revised results section. As a simplified example of what this means, consider when if a variable is identified as statically significant for $\alpha = 0.05$. We agree that this by itself is a relatively weak association. If the same variable is identified by three of six possible phenotypes (5 individual antibiotics, plus an MDR category), the probability is less ($0.05/3 = 0.017$) that this is a spurious association (we recognize that this probability example assumes independence, which is not likely to be the case, but it is why we chose to focus on variables that were consistently identified; and most identified variables are related to potential risk of transmission...)<<<

L78 Why assume that transmission is measured by colonisation and not give arguments whether that is indeed so. In that way you can assume anything and thus draw wrong conclusions. To make myself very clear, according to me: transmission is not measured by colonisation! Transmission is the transfer of (micro-)organism from one host (the infectious one) to the other host (the recipient one). Only by taking into account exposure one can estimate transmission. This can be done using for example by using $\#cases/\#susceptibles$ as dependent variable and putting $\#infectious/\#total$ number in the offset (Biemans, de Jong, and Bijma, 2017; Velthuis et al., 2002). Here you look only at recipient which can have very different exposure and even for wildlife exposure could also come from infectious individual that are treated with antibiotics.

>>>please see response to first comment. We don't disagree with the reviewer. Exposure is not measured directly in our study – but we can infer greater risk of transmission based on the fact that the livelihood variables that we identified can be reasonably assumed to be related to differential exposure and thus transmission<<<

Reviewer #2 (Remarks to the Author):

General Remarks. On initial reading this looked like an interesting article attempting to examine “one health AMR” in Tanzania - a LMIC region. The sample size is impressive and at first glance so is some of the analysis. However, the paper suffers from many serious deficiencies. Firstly, the sampling criteria whilst is balanced lacks rationale, for example, how were the subset of *E. coli* strain chosen for WGS?

>>> Please note that the WGS data was provided as an independent result that was consistent with the MLVA analysis. Isolates used in the MLVA analysis were selected randomly. For WGS, as noted in the original manuscript, isolates were from homes that were selected because of their inclusion in a separate network connectivity analysis (i.e., not random) and we have provided more detail on this selection in Lines 448-450. Nevertheless, the WGS analysis failed to detect significant partitioning of the *E. coli* population between hosts. This is consistent with the MLVA analysis and it is consistent with the broader narrative of widespread bacterial transmission in the study area. <<<

Secondly, assessing crude resistance is something that should be avoided as it does not differentiate intrinsic resistance from acquired and many of the bacteria on the screening plate we no doubt environmental bacteria or commensals. Therefore, it is very hard to ascribe any importance to the crude resistance data.

>>> To address concerns about our reliance on “crude resistance” and the possibility of confounded results due to intrinsic resistance, we compared the phenotype of the sequenced isolates with the antimicrobial-resistance genes identified from the sequence data. Our analysis, shows that our phenotypic testing provides 94-95% diagnostic specificity and 87-90% diagnostic sensitivity relative to the sequence data (which we treat as the “gold standard” for this comparison; this material is discussed on page 7 of the revised manuscript with frequency data provided in the new Table 1). The only antibiotic for which we had a consistent problem was streptomycin for which we probably picked a breakpoint that was too high (i.e., too many false negatives). The revised results section clarifies these points (see lines 116-127). We acknowledge the high likelihood that error is present in this analysis, but we have no a priori reason to assume that this error is nonrandom in distribution. If correct, there is no bias due to the error and the question becomes if the magnitude of error is too great to detect meaningful relationships. We argue that is not the case because of the consistent detection of livelihood factors that are related to transmission risk<<<

Thirdly, we are told that a subset of *E. coli* were subject to WGS yet the analysis we have is very superficial – what were the ARGs and did you look for virulence factors?

>>>please see our response above. The WGS data was meant to support the MLVA findings. We did include antibiotic-resistance genotypes in the original supplemental file (located there because the genotypes are not the focus of this manuscript). Importantly, further analysis of WGS data would not have any impact of the findings that are the focus of this manuscript. Nevertheless, for the revised manuscript we have included information with respect to diagnostic sensitivity and specificity (see above, Table 1 in the revised manuscript) and we moved the antibiotic-resistance genotype data from the supplemental file into the main text (Table 4).<<<

Fourthly, the mathematical modelling data is interesting but these to be found in the supplementary data only – in fact the choice of figures and tables for the main paper needs to be reassessed.

>>>We agree that these tables could provide more insight if they are included in the body of the manuscript, but this study relied on numerous analyses (MLVA, WGS, AST, logistic regression, etc.) and we prefer to keep the material in the main text as focused as possible. Including this information in the main text would add six tables. Given that we’ve already added two additional tables to this revision, we would appreciate the editor’s opinion about moving the large model tables into the main text as well<<<

Fifth, much of the analysis and narrative shows a degree of naivety with incorrect statements or conclusions that do align with the data – for example, the choice of antibiotics and not examining ESBLs is very strange. Also, the authors were surprised that samples showed resistance to ampicillin, streptomycin, sulfamethoxazole, tetracycline and trimethoprim – yet resistance mediated by all of these genes can be house in class 1 integrons which are very common.

>>>it is unclear what the reviewer means by “incorrect statements or conclusions.” The example given is not an incorrect statement or conclusion, but rather the reviewer’s personal preference that we would have tested for ESBL phenotypes. We argue that there is no *a priori* reason to expect a different outcome from adding another beta-lactamase phenotype. The distribution of the resistance phenotypes that we measured provided consistent relationships with variables that are related to risk of bacterial transmission. The second example given by the reviewer also seems to less than supportive of the claim that there are incorrect statements or with conclusions not aligning with the data. We point out that resistance to these antibiotics is higher by noting that “resistance to ampicillin, streptomycin, sulfamethoxazole, trimethoprim and tetracycline across livestock types was higher (>35% in Arusha, >30% in Maasai, and >10% in Chagga) compared to ceftazidime, chloramphenicol, ciprofloxacin and

kanamycin". We fail to see how this description conveys "surprise" at these findings nor is the role of class 1 integrons germane to the broader narrative about factors that predict carriage of antimicrobial-resistant bacteria<<<

In addition to the above, the use of English is careless with many incorrect sentences and typos throughout.

>>>We are not sure to what the reviewer is referring – no specific examples are given. We removed a duplicated sentence and endeavored to improve the writing but changes in this regard were fairly minimal. We concede that there were a number of typographical errors in the literature cited section, but these should be corrected with the revised manuscript.<<<

To conclude, the sample size is impressive but the analysis is poor and the study looks as though it was rushed.

Reviewer #3 (Remarks to the Author):

This is an interesting paper describing detailed analysis of enormous amount of information, collected in very interesting setting. I was not convinced with one of the major messages of the study: Antibiotic use does not contribute to prevalence of resistance, it is mostly livelihood factors (in the setting studied). While it is very convincing the livelihood factors are important, there is very little supporting evidence in the study that antibiotic use is not important as a facilitator of transmission in a high transmission setting. I suggest to tone down the statement regarding antibiotic effect, and just state that under the condition studied - no effect was found. You may want to discuss that this lack of effect may be due to close linearity with livelihood conditions, or because in high transmission setting this effect is difficult to ascertain, or because resistance to "old" antibiotics were studied, where the importance of antibiotic pressure has already reached its peak.

>>>as with reviewer 1, we agree that our title and language set inappropriate expectations about what was accomplished in this study. We did not measure transmission *per se*, but we identified factors that are consistent with increased risk of transmission. We also concur that we did not provide sufficient acknowledgement of the importance of antibiotic use as the mechanism responsible for emergence and proliferation of resistant strains. Our measures of antibiotic use for the present study may have been insufficient to detect a relationship. We modified the paper title and language in the introduction and discussion to better reflect these points.

Line 48, up to 48 isolates were picked from..., and latter average resistance per sample was calculated. - this results in an unconventional measure of resistance which is the product of how many samples had a resistant bacteria (among those picked randomly), and what was the abundance of resistant organisms in the sample. This measure and the interpretation needs to be discussed.

>>> Prevalence is calculated by dividing the positives by the total number tested. Assuming the isolates that were tested represent a random selection on agar plates that have no antibiotics (used during original isolation), we have no *a priori* reason to expect bias in this selection process and prevalence is what it is. We agree, however, that this method is not sensitive to rare phenotypes. We added a statement to the end of the first paragraph in the results to address this point. Nevertheless, the inability to detect rare phenotypes is not an issue for this analysis<<<

Analysis - of the similarity of mechanisms of resistance in isolates that were sequenced, will add an important dimension to the paper.

>>>The original submission included the resistance genotypes from the whole-genome sequence data (included in the supplemental file). We have expanded this point in the results including moving the table genotype table into the main text. We also added a correspondence analysis to quantify the degree of agreement between the phenotype and genotype (first section of results, plus table). Additional analysis of the sequence data (beyond the population genetics component that we included originally) would not

add additional insight to our conclusions so we did not expand the descriptive information from the whole-genome sequence data beyond these modifications.<<<

Specific comments:

line 51 repeats the same message as the sentence before. please revise. <<<done>>>

Table 2: please add the number of samples. <<<done>>>

Reviewers' Comments:

Reviewer #1:

Remarks to the Author:

I want to thank the authors for their rebuttal. I still do not agree with them but I can see that they understood my criticism, although we disagree how to weight it. I tried below to give suggestions how to further improve their interesting paper according to my opinion.

The paper has improved and I find it more acceptable. I think two points need to be addressed. Below I point to the places in the text where this could be done. Here I first explain the two points:

1. Transmission is important but here only the between-species transmission is (indirectly) studied. Other papers have called this spill-over or attribution. I think this should be mentioned that the research is only looking at the between-species transmission. I think it should also be mentioned that within-species transmission is important.
2. Separation between antibiotic use and transmission as factors determining the occurrence of ARM is not explained well. Antibiotic use may increase transmission but ARM may also transmit effectively without antibiotic use.

Remarks linked to line numbers:

Line 48: In this paragraph the transmission between species is addressed (as is clear from the second sentence) but nowhere the difference between within-species and between species transmission is discussed. In the high income countries transmission within-species could still be (probably is) important even when between-species is not.

Line 73: The assumption to identify risk factors: I still disagree with this wording. It is not conclusive evidence of transmission. Rather call it attribution although you know nothing about the possible direction of transmission.

Line 422: Why use spill-over here: maybe stay with the same wording everywhere: between-species transmission. Also consider not using transmission. Call it spill-over throughout. Others have used attribution. I think it is important that the basis for the conclusion and the fact that only between-species transmission is considered should be emphasized.

Line 428: Promotion of transmission by antibiotic use. Based on what ? Transmission without antibiotic use can be just as effective (both within and between species).

Mart C.M.de Jong

Reviewer #3:

Remarks to the Author:

I suggest to rephrase the conclusion statement: Our study suggests that prudent-use strategies should be given less priority in contexts where limitations in sanitation hygiene, and healthcare infrastructures give rise to high rates of bacterial transmission across communities

To a positive statement such as:

Our study suggests that higher priority should be given to improving sanitation hygiene, and healthcare infrastructures than to prudent-use strategies, especially in resources limited settings.

Reviewer #4:

Remarks to the Author:

This work is particularly impressive and interesting. It challenges the direct link between the prescription of manufactured antibiotics produced by humans and distributed to humans and animals and the level of bacterial resistance. As such, it is normal that some reviewers should be highly reluctant, particularly antibiotic specialists, since this goes against the simplistic model of the level of resistance directly proportional to the prescription of antibiotics. The only real criticism I can make, but probably through a bias or conflict of interest, is the lack of consideration of the dimension recently raised in several "Lancet Global Health" publications showing this general discrepancy in the world between the level of antibiotic production and distribution, and the level of resistance, in particular in *Escherichia coli*. This paper will be extremely cited, including by my team and corresponds to one of the major questions of our time which is: what is the source of the pressure of selection antibiotic resistance? This resistance is of natural origin, it can be transferred or can help to select populations that are naturally resistant or have acquired resistance to adapt to environmental pressure.

Reviewer 1

I want to thank the authors for their rebuttal. I still do not agree with them but I can see that they understood my criticism, although we disagree how to weight it. I tried below to give suggestions how to further improve their interesting paper according to my opinion.

The paper has improved and I find it more acceptable. I think two points need to be addressed. Below I point to the places in the text where this could be done. Here I first explain the two points:

1. Transmission is important but here only the between-species transmission is (indirectly) studied. Other papers have called this spill-over or attribution. I think this should be mentioned that the research is only looking at the between-species transmission. I think it should also be mentioned that within-species transmission is important.

>>>we thank the reviewer for willingness to work with the manuscript and to provide valuable insights. We'd like to clarify that we are drawing inferences based on the degree of phenotypic and genotypic similarity for bacterial isolates between host species, but our comparisons also include comparisons within the same species (e.g., genetic data presented in the ms shows no significant host-association and potential widespread dissemination within and between host species; several of the risk factors identified in this study involved animal markets, purchases, etc, which we infer to represent transmission between cattle). We revised the second paragraph of the introduction to further clarify the point that we are making inferences about transmission by characterizing phenotypes and genotypes within and between host species.

2. Separation between antibiotic use and transmission as factors determining the occurrence of ARM is not explained well. Antibiotic use may increase transmission but ARM may also transmit effectively without antibiotic use.

>>>we believe this point is made succinctly with the following sentence in the discussion: "That is, within these three communities, any *specific* effect of antibiotic use on the transmission of resistant bacteria appears to be overshadowed by livelihood factors that promote the transmission of bacteria *generally*."

Remarks linked to line numbers:

Line 48: In this paragraph the transmission between species is addressed (as is clear from the second sentence) but nowhere the difference between within-species and between species transmission is discussed. In the high income countries transmission within-species could still be (probably is) important even when between-species is not.

>>>see revised paragraph as discussed above

Line 73: The assumption to identify risk factors: I still disagree with this wording. It is not conclusive evidence of transmission. Rather call it attribution although you know nothing about the possible direction of transmission.

>>>we concur that finding similar distributions by itself does not necessarily represent transmission. For example, perhaps wildlife in Tanzania have always harbored antimicrobial-resistant enteric bacteria with a similar distribution as we find in other host populations, including people. If that was the case, no transmission is necessary to explain the distribution. We don't have historical data for comparison, but we are able to partition the variance in distribution amongst different risk factors, the majority of which are likely to be associated with transmission. Consequently, we argue that the assumption is reasonable.

Line 422: Why use spill-over here: maybe stay with the same wording everywhere: between-species transmission. Also consider not using transmission. Call it spill-over throughout. Others have used attribution. I think it is important that the basis for the conclusion and the fact that only between-species transmission is considered should be emphasized.

>>>we agree that introducing the term “spillover” in the concluding paragraph is not good form and we have revised and clarified the two sentences involved. As explained above, we disagree with the reviewer's assertion that only between-species transmission is considered.

Line 428: Promotion of transmission by antibiotic use. Based on what ? Transmission without antibiotic use can be just as effective (both within and between species).

>>>we agree completely, and this was the point of the sentence – we revised clarify.

Mart C.M.de Jong

Reviewer #3

I suggest to rephrase the conclusion statement: Our study suggests that prudent-use strategies should be given less priority in contexts where limitations in sanitation hygiene, and healthcare infrastructures give rise to high rates of bacterial transmission across communities

To a positive statement such as:

Our study suggests that higher priority should be given to improving sanitation hygiene, and healthcare infrastructures than to prudent-use strategies, especially in resources limited settings.

>>>agreed and changed accordingly

Reviewer 4

This work is particularly impressive and interesting. It challenges the direct link between the prescription of manufactured antibiotics produced by humans and distributed to humans and animals and the level of bacterial resistance. As such, it is normal that some reviewers should be highly reluctant, particularly antibiotic specialists, since this goes against the simplistic model of the level of resistance directly proportional to the prescription of antibiotics. The only real criticism I can make, but probably through a bias or conflict of interest, is the lack of consideration of the dimension recently raised in several “Lancet Global Health” publications showing this general discrepancy in the world between the level of antibiotic production and distribution, and the level of resistance, in particular in *Escherichia coli*. This paper will be

extremely cited, including by my team and corresponds to one of the major questions of our time which is: what is the source of the pressure of selection antibiotic resistance? This resistance is of natural origin, it can be transferred or can help to select populations that are naturally resistant or have acquired resistance to adapt to environmental pressure.

>>>thank you for these positive comments. We believe that our findings fully support the global-scale associations that have been recently described in the literature, particularly by Collignon et al. 2018 (which is cited in our manuscript). We would also argue that our findings demonstrate that the published “macro-level” observations scale directly to the local level (although the local level analysis provides important context that can guide specific interventions and investments...something that is much more difficult to accomplish with macro-level observations).